# Architecture of the U6 snRNP reveals specific recognition of 3′-end processed U6 snRNA

Eric J. Montemayor[1,2], Allison L. Didychuk[1], Allyson D. Yake[2], Gurnimrat K. Sidhu[1], David A. Brow[2] & Samuel E. Butcher [1]

The spliceosome removes introns from precursor messenger RNA (pre-mRNA) to produce mature mRNA. Prior to catalysis, spliceosomes are assembled de novo onto pre-mRNA substrates. During this assembly process, U6 small nuclear RNA (snRNA) undergoes extensive structural remodeling. The early stages of this remodeling process are chaperoned by U6 snRNP proteins Prp24 and the Lsm2–8 heteroheptameric ring. We now report a structure of the U6 snRNP from *Saccharomyces cerevisiae*. The structure reveals protein–protein contacts that position Lsm2–8 in close proximity to the chaperone "active site" of Prp24. The structure also shows how the Lsm2–8 ring specifically recognizes U6 snRNA that has been post-transcriptionally modified at its 3′ end, thereby elucidating the mechanism by which U6 snRNPs selectively recruit 3′ end-processed U6 snRNA into spliceosomes. Additionally, the structure reveals unanticipated homology between the C-terminal regions of Lsm8 and the cytoplasmic Lsm1 protein involved in mRNA decay.

[1] Department of Biochemistry, University of Wisconsin-Madison, Madison, WI 53706, USA. [2] Department of Biomolecular Chemistry, University of Wisconsin-Madison, Madison, WI 53706, USA. Correspondence and requests for materials should be addressed to E.J.M. (email: emontemayor@wisc.edu) or to D.A.B. (email: dabrow@wisc.edu) or to S.E.B. (email: sebutcher@wisc.edu)

Eukaryotes are unique from bacteria and archaea in that they employ a massive and highly dynamic ribonucleoprotein assembly called the spliceosome to remove introns from precursor mRNA (pre-mRNA) before its translation into protein[1–4]. This process likely arose to protect the transcriptome from invasive genetic elements[5], but evolved to expand the coding potential of the genome and afford layers of gene regulation that are a hallmark of eukaryotic life[6]. The spliceosome contains five small nuclear RNAs (snRNAs U1, U2, U4, U5, and U6) and more than a hundred protein cofactors[7,8] that are recruited to introns via protein–protein, protein–RNA, and RNA–RNA interactions[9]. Spliceosome assembly, catalytic activation, and disassembly also require the function of at least eight ATP-dependent helicases belonging to the DEAD-box, DEAH/RHA, and Ski-like protein superfamilies, which unwind or remodel snRNA–snRNA, snRNA–intron, and snRNA–exon duplexes[10]. U6 snRNA is unique among the spliceosomal RNAs in that its internal stem loop (ISL) must be transiently unwound and annealed to U4 snRNA during spliceosome assembly[11] by Prp24, an RNA chaperone lacking homology to the ATP-dependent helicases. Thus, U6 recruitment into spliceosomes proceeds though a novel, and poorly understood, RNA remodeling mechanism involving large-scale conformational changes.

The *Saccharomyces cerevisiae* U6 small nuclear ribonucleoprotein (snRNP) contains U6 snRNA, Prp24, and the Lsm2–8 heteroheptameric ring[12–16] (Supplementary Fig. 1). Prp24 was first identified as a U6 remodeling factor when mutations in the yeast protein were found to suppress cold-sensitive growth arising from destabilization of U4/U6 di-snRNA[13]. Subsequent in vitro biochemical and structural analysis of Prp24 identified specific binding sites between the protein and U4/U6[17,18], and free U6[19–23]. Recent work identified an "electropositive groove" along the surface of Prp24 that does not contact U6 in the U6 snRNP[22–24], but plays an important role in U4/U6 assembly[24]. As such, the "electropositive groove" likely represents the active site of Prp24 during the annealing of U6 and U4.

The Lsm2–8 heteroheptameric ring cooperatively binds U6 snRNA and accelerates Prp24-mediated annealing of U4/U6 di-snRNA though an unknown mechanism[24–27]. One possible explanation for this activity arises from homology between the Lsm ring and the Hfq protein families[28], which bind and remodel RNAs though numerous "proximal," "distal," and "rim" binding sites that place cognate RNAs in close proximity to one another[29,30]. However, the spatial organization of the Lsm2–8 ring in the U6 snRNP, particularly with respect to the electropositive groove of Prp24, could not be determined in prior electron microscopy reconstructions of the yeast U6 snRNP[16].

In addition to stimulating ATP-independent remodeling of U6 snRNA, the Lsm2–8 ring specifically recognizes the 3′ end of U6 RNA, with a preference for binding RNA that has been post-transcriptionally modified by the 3′ exoribonuclease Usb1, which leaves a 3′ terminal phosphate on U6[31–35]. Prior crystal structures of *S. cerevisiae* Lsm2–8 lacked the complete C-terminus of Lsm8, which is known to control nuclear localization of the Lsm ring[36], and employed RNAs terminating with a 2′,3′-cis diol that do not resemble the modified form of U6 found in vivo[31,35,37,38].

Despite a recent surge in the pace of spliceosome structure determination by cryo-electron microscopy, the mechanisms underlying structural rearrangements in spliceosomes and their constituent snRNPs remain poorly understood[39,40]. In order to provide mechanistic insight into an early step of U4/U6.U5 tri-snRNP assembly, we determined crystal structures of U6 snRNPs from *S. cerevisiae*, containing U6 snRNA, Prp24, and the Lsm2–8 heteroheptameric ring. The structures show that the Lsm2–8 ring is positioned in close proximity to the active site of Prp24 and suggest a possible mechanism for remodeling of RNA.

We also show that the 3′ end of mature, post-transcriptionally processed U6 binds the Lsm2–8 ring in a different register relative to non-processed U6, and use in vitro biochemistry to dissect how Lsm2–8 specifically targets post-transcriptionally processed U6 RNA for inclusion into spliceosomes. The structure also reveals unanticipated homology between the C-terminal regions of Lsm8 and the cytoplasmic Lsm1 protein involved in mRNA decay[41,42].

## Results

**Crystal structure of the U6 snRNP.** We reconstituted the U6 snRNP from *S. cerevisiae*, using in vitro transcribed U6 snRNA and recombinant Prp24 and Lsm2–8. The complex lacks the poorly-conserved C-terminus of Lsm4 and the 5′ stem of U6, the latter of which is not necessary for efficient U6 snRNP formation or U4/U6 annealing[24] (Fig. 1, Supplementary Table 1 and Supplementary Figs. 2, 3, Supplementary Data 1, Supplementary Note 1). The core region of the U6 snRNP retains the unique "interlocked rings" topology of Prp24 and U6 snRNA and is virtually identical to previously determined structures of U6/Prp24 binary complexes[22,23], with a comparative r.m.s.d of ~1.3 Å. There are protein–protein contacts in the U6 snRNP that were not visible in previous structures, including a 660 Å$^2$ contact between RRM4 of Prp24 and Lsm2, and a contact between the conserved C-terminus (or "SNFFL box") of Prp24 and Lsm5 and Lsm7. Together, these contacts position the Lsm2–8 ring in close proximity to the electropositive groove on Prp24 that had been shown previously to function in annealing of U4 and U6 snRNAs[24] (Fig. 1d, Supplementary Movie 1).

The structure of the Lsm2–8 ring in the U6 snRNP is similar to previously determined structures of Lsm rings alone bound to short oligonucleotides[43,44]. However, the C-terminal region of Lsm8 that was absent in the previous Lsm2–8 structure is visible here. It spans the U6-distal face of the ring, partially capping the central hole and forming an alpha helix that is anchored at its C-terminus by contacts with Lsm3 and Lsm6 (Fig. 2a, b). The C-terminal region of Lsm1 in the Lsm1–7 ring also crosses the distal face of the Lsm ring[44], revealing a common architecture of the Lsm1–7 and Lsm2–8 rings (Fig. 2c, d). Interestingly, the Pat1 binding site on the Lsm1–7 ring, comprising Lsm2 and Lsm3[44], is almost entirely exposed in the U6 snRNP (Fig. 2e). Thus, the observed architecture of the U6 snRNP may be compatible with binding of Pat1. Indeed, a complex containing Pat1, U6, Lsm2–8, and Prp24 was recently detected in human nuclei[45].

**Functional intermolecular contacts in the U6 snRNP.** The observed contact between the highly conserved SNFFL box C-terminus of Prp24 and Lsm5 and Lsm7 (Fig. 3a, b) is consistent with previously reported yeast two-hybrid interactions[25]. The linker between the SNFFL box and RRM4 spans ~30 residues and is disordered. The linker lacks sequence conservation, but is well conserved in length and as such likely constrains the orientation of the Lsm2–8 ring relative to the last RRM domain in Prp24. In order to test the functional importance of the additional and unanticipated contact between Prp24 RRM4 and Lsm2 (Fig. 3c), we substituted eight RRM4 residues at the interface with alanine ("Prp24-8Asub") and assayed U6 and Lsm2–8 binding affinity in vitro. Prp24-8Asub had unaltered affinity for U6 snRNA (Fig. 3d), but reduced the affinity of U6•Prp24 for Lsm2–8 two-fold (Fig. 3e). Additionally, a two-fold slower rate of U4/U6 annealing was observed when Prp24-8Asub was used in place of wild-type Prp24 (Fig. 3f). Therefore, the Prp24 RRM4-Lsm2 interaction stabilizes the U6 snRNP and helps promote annealing of U4/U6 RNA in vitro. In vivo, *prp24-8Asub* displays a heat-sensitive growth phenotype (Fig. 3g and Supplementary Table 2). Combining the Prp24-8Asub and ΔSNFFL mutations

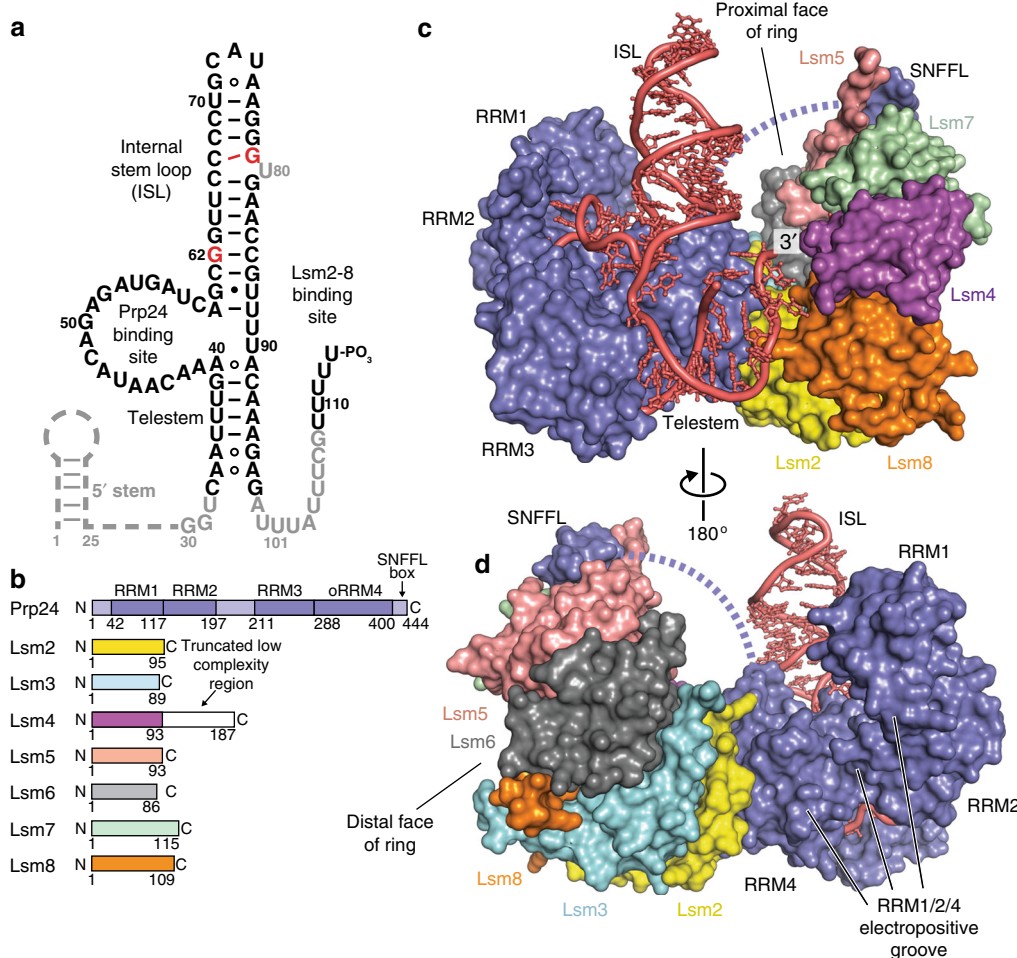

**Fig. 1** Architecture of the U6 snRNP from *Saccharomyces cerevisiae*. **a** Sequence and observed secondary structure of the U6 snRNA used for crystallization. Disordered nucleotides are shown in gray text, and the absent 5′ stem region is shown as a dashed line. The 30–113 nucleotide RNA that was used for crystallization has a 3′ phosphate group and A62G/A79G mutations (colored red) to quench structural dynamics in the U6 internal stem loop (ISL), which do not alter the core topology of the U6/Prp24 complex[22,23]. The 30–112 RNA that yielded a different crystal form contains a 2′ phosphate group and lacks the A79G mutation. **b** Primary structure of protein components in U6 snRNP samples used for crystallization. The occluded (o) RRM4 of Prp24 has additional N- and C-terminal alpha helices appended to its RRM core[76], and is denoted as RRM4 hereafter. In PDB 5VSU, Prp24 residues 1–26 and 399–431 are disordered. In PDB 6ASO, Prp24 residues 1–27 are truncated; 28–29 and 399–444 are disordered. The C-terminus of Lsm4 is predicted to be disordered and was therefore truncated from the crystallization construct (Supplementary Note 1). **c** Overall architecture of the yeast U6 snRNP, as observed in two distinct crystal structures (Supplementary Table 1). U6 snRNA (red) and Prp24 (blue) contact the "proximal" face of the Lsm2-8 ring. **d** Alternate view of the U6 snRNP. The ~ 30-residue linker region between RRM4 and the C-terminal SNFFL box motif of Prp24 is disordered and depicted as a dashed line

resulted in even slower growth at 37 °C than the 8Asub mutation alone (Fig. 2g). Together, these findings show that the observed architecture of the U6 snRNP likely represents an on-path species in Prp24-mediated annealing of U4/U6 di-snRNPs, where placement of the Lsm2–8 ring in close proximity to the electropositive groove of Prp24 is partly, but not entirely, responsible for enhanced annealing in the presence of Lsm2–8.

The observed protein–protein contacts in the U6 snRNP are compatible with binding of either the Lsm1–7 or Lsm2–8 rings in U6 snRNPs. However, Lsm1 lacks the non-consensus nuclear localization sequence found in Lsm8[36], and therefore cytoplasmic Lsm1–7 rings are unlikely to be recruited into U6 snRNPs.

**3′ end recognition of mature U6 snRNA.** The yeast Lsm2–8 ring preferentially binds U6 snRNAs with a non-cyclic 3′ phosphate, a post-transcriptional modification installed by the exoribonuclease Usb1 [31]. In order to elucidate the basis for this selectivity, we generated U6 snRNPs with 3′ terminal phosphate groups[31,37].

We also used RNAs with either four or five 3′ uridines to probe the structural effect of 3′ length heterogeneity inherent to *S. cerevisiae* U6 snRNA[12].

The 3′ end of U6 binds to the central cavity of Lsm2–8 but is not threaded completely through it (Fig. 2a, b, Supplementary Movie 1). Nucleotides between the telestem and 3′ oligouridylate tail are poorly resolved due to an apparent lack of protein-RNA contacts, which likely affords a degree of flexibility to accommodate the 3′ length heterogeneity of U6 snRNA[12]. Inside the ring, RNA is bound in a shifted register relative to that observed previously (Fig. 4a–c)[38]. Instead of being bound in the Sm-like pocket of Lsm3, the terminal nucleotide occupies a new pocket comprised of residues in Lsm2, Lsm3 and Lsm8 (Fig. 4d, e), and the penultimate nucleotide is bound in the previously identified Sm-like pocket of Lsm3. In this new "Up" (or uridine phosphate) pocket, Lsm2 residue K20 forms a cation–pi interaction with the terminal uracil and a buried salt bridge with Lsm3 residues D71 and R69, the latter of which participates in a cation–pi interaction with the penultimate uracil. The remaining contacts with the

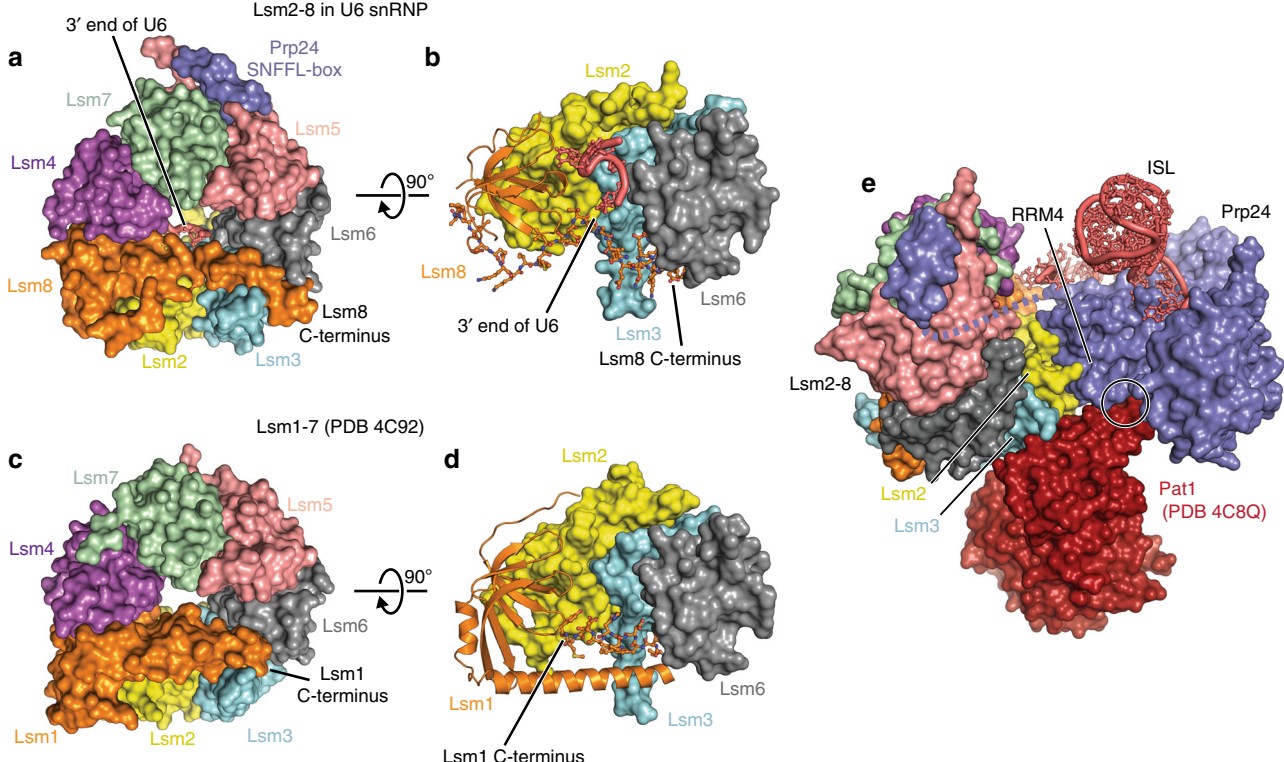

**Fig. 2** Comparison of the Lsm2–8 and Lsm1–7 rings. **a** The C-terminal region of Lsm8 spans the distal face of the Lsm2–8 ring, contacts the 3′ end of U6 snRNA inside the ring, and forms a C-terminal alpha helix that contacts Lsm3 and Lsm6. **b** Detailed view of Lsm8 and the 3′ end of U6 snRNA. Lsm4, Lsm5, and Lsm7 are hidden for clarity. Lsm8 is shown in cartoon, except for the C-terminal 34 residues which are shown in ball and sticks. **c** The C-terminal region of Lsm1 also crosses the distal face of the Lsm1–7 ring via a longer alpha helix that terminates between Lsm3 and Lsm6 before being threaded back into the interior of the ring. **d** Detailed view of Lsm1 in the Lsm1–7 ring, depicted as in panel **c**. The C-terminal 12 residues of Lsm1 are shown in ball and sticks. **e** Superposition of Lsm2–8 in the U6 snRNP with Lsm1–7 bound to RNA decay factor Pat1. The Pat1 binding site on Lsm2 and Lsm3 is solvent exposed in U6 snRNPs. With the exception of a small steric clash between RRM4 and Pat1 (denoted by a circle), the architecture of the U6 snRNP appears to be compatible with binding of Pat1 to Lsm2 and Lsm3

terminal uridylate are dominated by electrostatic interactions with Lsm3-R21 and Lsm8-K87, K90 and K92. The importance of electrostatics in the center of the Lsm2–8 ring is evinced by near identical binding mechanisms when the phosphate is attached to either the 2′ or 3′ oxygen of the terminal uridine (Fig. 4a, b). Additionally, the binding register is unchanged relative to the terminal nucleotide by the number of uridines at the 3′ end of U6.

We tested the importance of these 3′ end recognition elements in vivo. Mutation of Lsm2 residue K20 to either alanine or glutamate was lethal (Supplementary Fig. 4). This phenotype likely arises from reduced association of Lsm2–8 with U6 snRNA, as in vitro binding assays show a large decrease in binding affinity (Supplementary Table 3). Mutation of Lsm3 residue R21 to alanine did not yield a discernible growth phenotype, but a charge inversion of arginine to aspartate results in both cold and heat sensitive growth (Supplementary Fig. 4a), consistent with a more severe binding defect in vitro (Supplementary Table 3). Substitution of Lsm8 residues 87–92 with alanines ("Lsm8-6Asub") did not display a growth defect, but deletion of the entire C-terminus of Lsm8 resulted in a lethal phenotype that likely arises from ablation of the non-consensus nuclear localization sequence in Lsm8[36] (Supplementary Fig. 4c). These mutations also reduced in vitro binding affinity between the U6 3′-end and Lsm2–8, but to a lesser extent than the lethal Lsm2 K20 mutation. Interestingly, all the above mutations invert Lsm2–8's binding preference for 3′-end processed U6 snRNA, with the mutants preferentially binding to a non-phosphorylated 3′ end (Fig. 5 and Supplementary Table 3). Thus, the 3′ binding pocket observed

here defines the structural basis for preferential incorporation of 3′-end processed U6 snRNA into spliceosomes.

## Discussion

U6 snRNA is associated with Lsm2–8 along the ring's interior and proximal face. As U6 is not threaded through the ring, it does not contact the distal face of Lsm2–8. This utilization of "end recognition" of RNA in the interior of Lsm2–8 is distinct from that of the homologous Sm and Hfq ring proteins, which bind RNA at both the proximal, interior, and distal faces of the ring[28–30,46–48] (Supplementary Fig. 5a). Thus, Lsm2–8 binds U6 in a manner that exposes known distal RNA binding surfaces in homologous ring structures. This architecture may promote additional contacts with the U4 snRNP that help to recruit U4 snRNA to the electropositive groove (Fig. 1d), in accordance with surface conservation in the Lsm2–8 ring that is not readily attributed to known binding surfaces for Prp24, Pat1, or U6 RNA[44] (Supplementary Fig. 5b, c).

The architecture of the free U6 snRNP provides insights into how spliceosomal RNAs can be remodeled in the absence of an ATP-dependent helicase. Instead of active disruption of duplex RNAs[49], components of the U6 snRNP appear to define a specific path towards annealing of U4 and U6 by providing a complementary surface for binding of "on path" secondary structure elements, such as duplex RNA in the electropositive groove of Prp24[22,24]. The architecture of the U6 snRNP may also prevent formation of trapped annealing intermediates. For

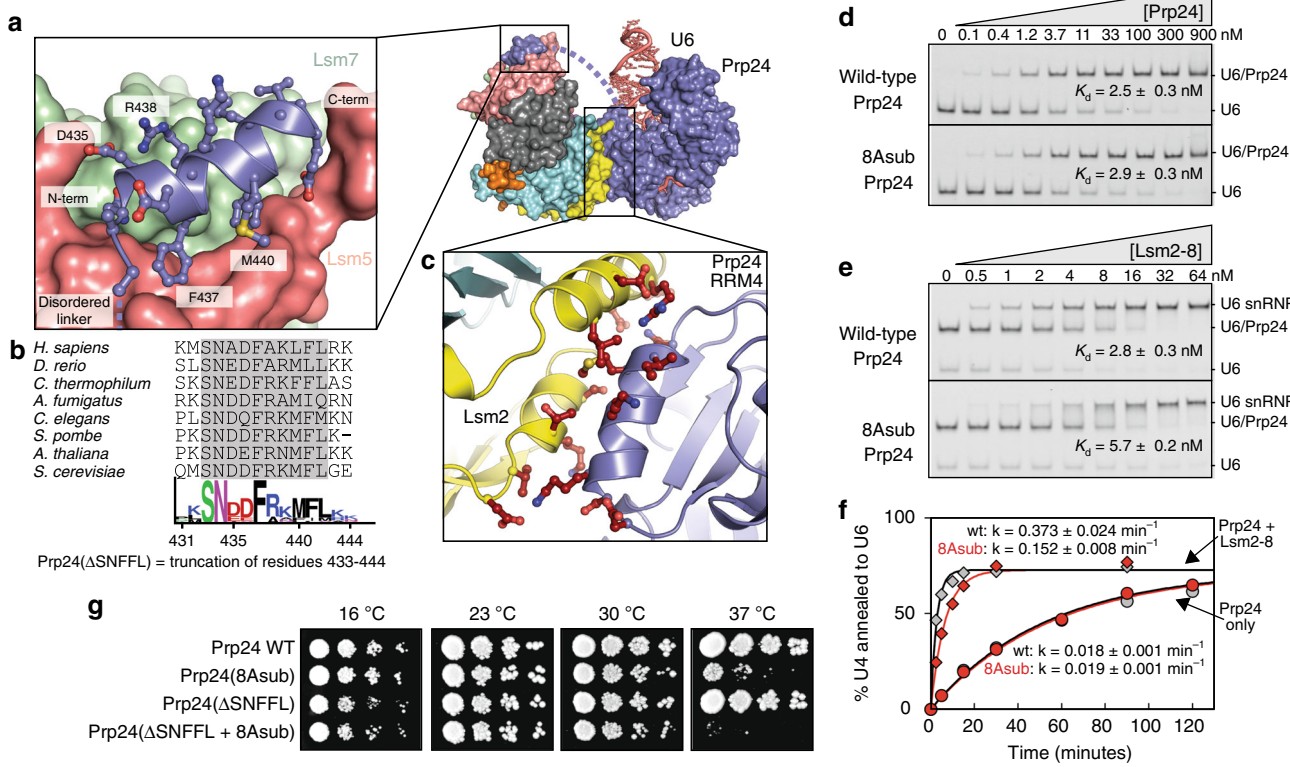

**Fig. 3** Prp24-Lsm2-8 ring contacts in the U6 snRNP. **a** Binding of the C-terminal SNFFL box motif of Prp24 to Lsm5 and Lsm7. **b** Sequence conservation of the C-terminal "SNFFL box" motif in Prp24. Numbering below the MEME consensus sequence corresponds to *S. cerevisiae* Prp24. **c** An unexpected contact between RRM4 of Prp24 and Lsm2. Prp24 residues depicted in red (D361, S362, K363, K367, L369, M370, I371, and N373) were mutated to alanine in the 8Asub allele. Lsm2 residues depicted in red were mutated as described in Supplementary Fig. 4. **d** Native PAGE with labeled U6 shows that substitution of Prp24 residues at the RRM4-Lsm2 interface (8Asub) does not affect U6 binding, as expected from the known U6/Prp24 interface[22,23]. **e** In contrast, Prp24-8Asub reduces Lsm2-8 affinity for U6/Prp24 two-fold and, **f** reduces the rate of in vitro annealing of U4 and U6 two-fold. Depicted binding and annealing data points represent the average of three technical replicates. **g** Deletion of the SNFFL box enhances the heat-sensitivity of a *prp24-8Asub* strain

example, protein–protein contacts between RRM4 of Prp24 and Lsm2–8 ring likely restrict the length, and thus stability, of the U6 telestem (Supplementary Fig. 1a). As such, the telestem is poised to unwind during annealing and therefore resolve the interlocked core architecture, so that Prp24 may subsequently be displaced from U4/U6. Nonetheless, further work is still required to determine the exact mechanism of U4/U6 annealing, particularly the importance of localizing the Lsm2–8 ring to the electro-positive groove of Prp24. Structure determination of a trapped U4/U6 annealing intermediate prior to displacement of Prp24 would likely provide such insights.

We note apparent co-variation of the C-terminus of Lsm8 and the 3′ end of mature U6 snRNA (Fig. 6 and Supplementary Note 1). In vertebrates, plants, and the fission yeast *Schizosaccharomyces pombe*, the C-terminus of Lsm8 is highly conserved in both sequence and length, ending with a conserved histidine residue that, via sequence alignment, overlaps with *S. cerevisiae* residues that contact the terminal uridine. The C-terminal histidine of Lsm8 in these organisms could conceivably interact with the terminal uridine base in U6 snRNA or the 2′,3′-cyclic phosphate group found on metazoan U6 RNAs[37]. We speculate that the sequence composition of Lsm8's C-terminus can be used to predict corresponding 3′ end chemistry in U6 snRNA. For example, we propose the *Saccharomycetaceae* orthologs of U6 shown in Fig. 6 (*N. diarenensis*, *Z. rouxii*, and *L. fermentatii*) harbor non-cyclic phosphates. By extension, *Saccharomycetaceae* orthologs of Usb1 are expected to harbor cyclic nucleotide phosphodiesterase activity, unlike human Usb1[31,32].

Free Lsm1–7 lacks robust binding affinity for RNA despite pronounced structural similarity between the nucleotide binding pockets of Lsm1 and Lsm8[38,43,44] (Fig. 4e). In light of the important role of the C-terminus of Lsm8 observed here, we hypothesize that the C-terminal region of Lsm1 (Fig. 2d) inhibits binding of RNA to Lsm1–7. Since association of Pat1 with Lsm1–7 restores binding affinity for RNA[50], we further speculate that Pat1 acts as an allosteric regulator of Lsm1–7, stimulating RNA binding activity via displacement of the Lsm1 C-terminus from the distal face of the ring. Further work is required to identify the exact sequence of RNA that associates with Lsm1–7 and the structure of the corresponding complex.

## Methods

**Prp24 expression and purification.** *Escherichia coli* BL21(DE3) STAR pLysS cells (Invitrogen) harboring a modified pET plasmid (Novagen) were used to produce *S. cerevisiae* Prp24 with a non-cleavable polyhistidine tag appended to the C-terminus of the protein. The Prp24 protein used in the 3′ phosphate-terminated U6 snRNP complex (PDB 6ASO) lacked the N-terminal 26 residues of Prp24, as the corresponding residues were disordered in the 2′ phosphate-terminated U6 snRNP with an intact N-terminus of Prp24 (PDB 5VSU). Cells were grown in Terrific Broth at 37 °C with shaking to an OD$_{600}$ of ~2, placed on ice for 1 h, and protein expression subsequently induced by addition of 1 mM isopropyl β-D-1-thiogalactopyranoside (IPTG) and additional growth with shaking at 16 °C for 20 h. Cells were harvested by centrifugation at 4000 × *g* for 10 min. Cell pellets from 1 L of culture were resuspended in 30 mL immobilized metal affinity chromatography (IMAC) buffer (500 mM NaCl, 50 mM HEPES acid, 50 mM imidazole base, 10% (v/v) glycerol, 1 mM tris(2-carboxyethyl)phosphine (TCEP) HCl). Lysozyme (Fisher), DNase I (Sigma), and protease inhibitors (EMD Millipore) were added prior to lysis of the cells via one freeze-thaw cycle and sonication. The resulting lysate was clarified by centrifugation at 48,000 × *g* for 30 min. The soluble fraction was loaded onto a 20 mL Ni$^{2+}$-charged nitrilotriacetic acid (NTA) agarose resin (Qiagen) that had been

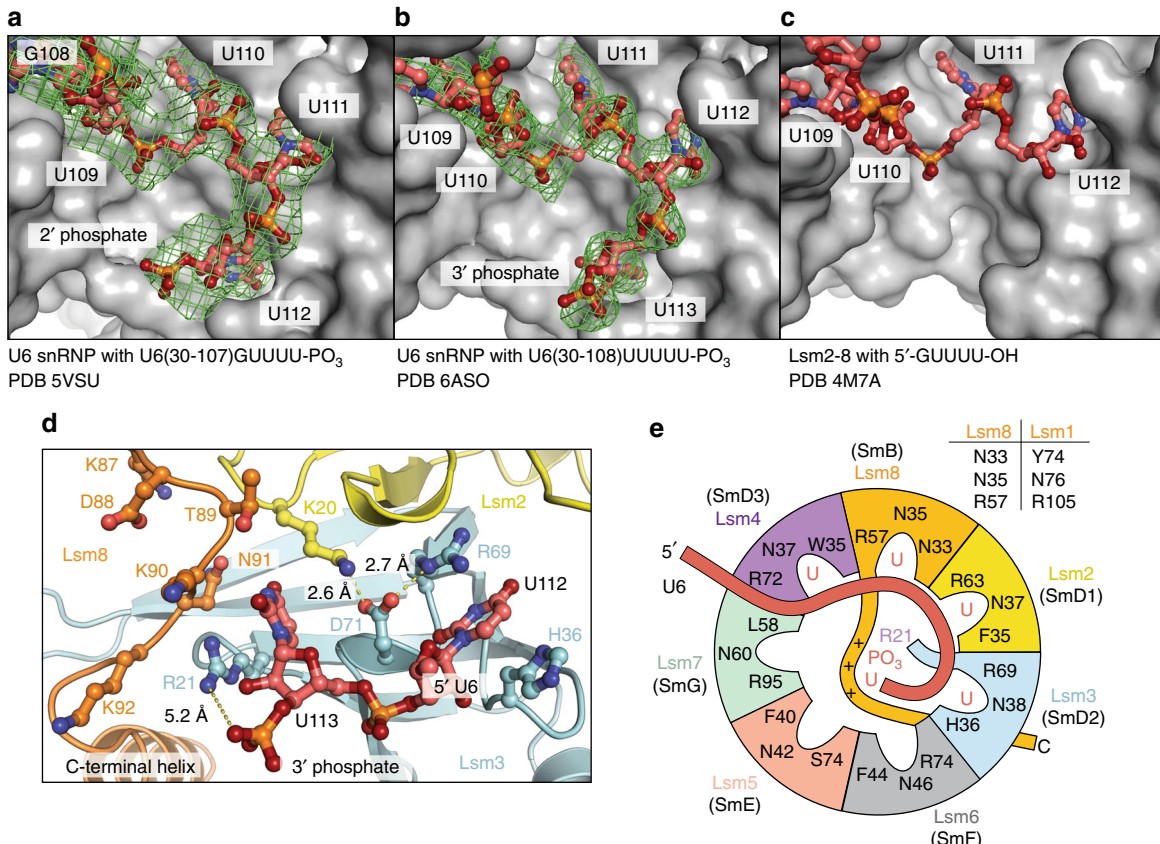

**Fig. 4** 3′ end recognition of mature U6 RNA by Lsm2–8 in *S. cerevisiae*. Comparison of unprocessed U6 and 3′ phosphorylated U6 binding mechanisms. Annealed omit maps ($mF_o$-D$F_c$, 2.5 r.m.s.d.) showing electron density for the phosphorylated U6 ends observed in U6 snRNPs. Lsm5 and Lsm6 are omitted for clarity in panels **a**–**c**. **a** In U6 snRNPs with four terminal uridines and a terminal 2′-phosphate, the last nucleotide occupies a 2′- or 3′-uridylate phosphate (Up) binding pocket in the center of the Lsm2-8 ring. **b** A five-uridine 3′ tail with a terminal 3′-phosphate shifts the binding register to accommodate the terminal nucleotide in the Up pocket. **c** A previously reported structure of Lsm2-8 bound to a short oligonucleotide with a four-uridine 3′-tail and 2′,3′-*cis* diol[37] has the same register as the five-uridine 3′ tail with a terminal 3′-phosphate, leaving the Up pocket empty. **d** Detailed view of the Up pocket in the central cavity of Lsm2-8. The terminal 3′-uridylate contacts Lsm2-K20, Lsm3-R21, and the C-terminus of Lsm8. The Lsm3-R21 distance from the terminal phosphate is beyond that for hydrogen bonding, and likely forms a long-range electrostatic interaction, which could be enhanced by the helical dipole in the C-terminal region of Lsm8. **e** Schematic of U6 3′-end binding by the Lsm2–8 ring. With the exception of Lsm5 and Lsm7, all Lsm proteins harbor canonical Sm-like binding motifs involving planar, cationic, and asparagine residues. Homologous Sm proteins are in parentheses. Residue numbering corresponds to *S. cerevisiae*. Equivalent residues in the putative nucleotide binding pocket of Lsm1, which replaces Lsm8 in the Lsm1–7 ring, are shown beside the cartoon

pre-equilibrated with IMAC buffer. The column was washed with 50 mL fresh IMAC buffer and Prp24 desorbed using IMAC buffer supplemented with 500 mM imidazole pH 7.0. The purified protein was dialyzed overnight at 4 °C against 1 L of ion exchange chromatography buffer (100 mM NaCl, 10 mM HEPES acid, 10 mM sodium HEPES, 10% glycerol, 1 mM TCEP HCl, 1 mM sodium azide, pH 7.0), and further purified using cation exchange chromatography with salt gradient elution on an AKTA FPLC system equipped with a 5 mL HiTrap heparin column (GE Healthcare). Protein concentration was estimated from UV absorbance and anticipated extinction coefficients at 280 nm[51]. Protein was stored at 4 °C. Protein stocks that were used for electrophoretic mobility shift assays differed from that above by use of a modified pET plasmid encoding a TEV-labile N-terminal polyhistidine tag, which was removed by addition of 1 mg TEV protease during dialysis prior to ion exchange purification.

**Lsm2-8 expression and purification.** Plasmid pQLink-Lsm2–8, which contains independent expression cassettes for all seven Lsm2–8 proteins, was used to synthesize recombinant *S. cerevisiae* Lsm2–8 with an N-terminal histidine tag on Lsm8, an N-terminal glutathione S-transferases (GST) tag on Lsm6, and a truncated variant of Lsm4 lacking the C-terminal 94 amino acids (henceforth referred to as "mostly wild type")[38,52]. The plasmid was transformed into *E. coli* BL21(DE3) STAR pLysS cells (Invitrogen), which were grown, induced, and harvested, and Lsm2–8 purified via IMAC, as above for Prp24. After collecting the IMAC eluate, the protein was dialyzed overnight into ion exchange buffer at 4 °C. Centrifugation was used to remove the large amount of precipitate that formed during dialysis. Soluble protein was further purified via glutathione agarose chromatography

(GenScript), with elution in ion exchange buffer supplemented with 50 mM HEPES acid, 50 mM sodium HEPES, 10 mM reduced glutathione. One milligram of TEV protease was added to the eluate, which was then dialyzed overnight at room temperature against 1 L of fresh ion exchange buffer. Soluble protein was purified on an FLPC system as above with a 5 mL HiTrap Q column (GE Healthcare). The eluate was diluted two-fold with one volume of water and then further purified with a 5 mL HiTrap heparin column (GE Healthcare) as above for Prp24. Peak fractions were collected and stored at 4 °C. The resulting "mostly wild type" protein complex was used for crystallization of the 2′ phosphate-terminated U6 snRNP (PDB 5VSU) and electrophoretic mobility shift assays.

A modified Lsm2–8 construct (called "construct 2") was used for fluorescence polarization binding experiments and crystallization of the 3′ phosphate-terminated U6 snRNP (6ASO), which differed from the above "mostly wild type" construct by a TEV-labile maltose binding protein (MBP) tag on Lsm6, a non-cleavable C-terminal hexahistidine tag on Lsm8 and a non-cleavable C-terminal StrepII tag on Lsm3, and deletion of the non-native N-terminal Met-Gly-Ser tripeptides from Lsm2, Lsm3, Lsm4, Lsm5, and Lsm7. Purification of this ring utilized amylose agarose (New England Biolabs) instead of glutathione agarose chromatography, and Strep-Tactin sepharose chromatography (IBA Lifesciences) was utilized immediately prior to HiTrapQ chromatography (GE Healthcare). For both Lsm2–8 preparations, protein concentration was estimated from UV absorbance and anticipated extinction coefficients at 280 nm[51]. Protein was stored at 4 °C.

**Mutant YfcE protein expression and purification.** A mutant variant of *E. coli* protein YfcE (C74H)[53] was used to prepare U6 nucleotides 30–113 with a

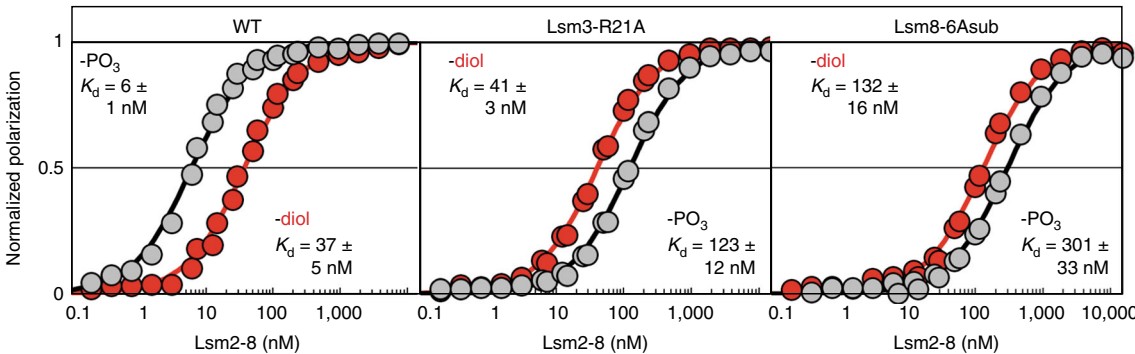

**Fig. 5** The terminal nucleotide binding pocket dictates preferential incorporation of mature U6 snRNA into spliceosomes. In vitro binding affinity between the 3′ end of U6 and Lsm2–8, as determined by fluorescence polarization. Mutation of elements that interact with the terminal phosphate invert Lsm2-8's binding preference for mature U6. The Lsm8-6Asub construct harbors six alanine substitutions spanning residues 87–92. Depicted data are from two technical replicates. Additional binding data are summarized in Supplementary Table 3

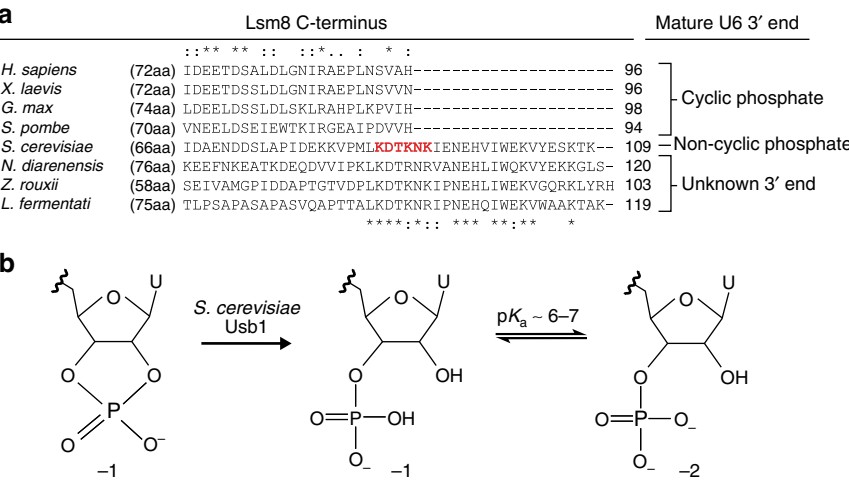

**Fig. 6** Covariation of U6 3′ end structure and the C-terminus of Lsm8. **a** Sequence conservation among the top four sequences is shown above the alignment, while conservation for the remaining sequences is below the alignment. The residues depicted in red were mutated to alanine in Fig. 5, Supplementary Fig. 4, and Supplementary Table 3. Organisms in which U6 has a 2′,3′-cyclic phosphate terminus lack a C-terminal extension of Lsm8[32,33,37,77]. In contrast, numerous *Saccharomycetaceae* orthologs of Lsm8 resemble *S. cerevisiae*, which harbors a non-cyclic phosphate on U6[31,37]. **b** Cyclic phosphorylated U6 has a charge of −1 while non-cyclic phosphorylated U6 can have a charge of −1 or −2, due to the latter's second ionization constant that is close to physiological pH[78]. The longer C-terminal region of *S. cerevisiae* Lsm8, with conserved lysine or arginine residues, likely compensates for the greater negative charge on the 3′ end of *S. cerevisiae* U6 snRNA

3′ non-cyclic phosphate, exploiting the enzyme's engineered 2′,3′-cyclic nucleotide 2′-phosphodiesterase activity and the cyclic phosphate moiety on U6 that remains after Hepatitis delta virus (HDV) cleavage (see RNA production section below). A codon-optimized variant of mutant YfcE was cloned into a modified pET vector and protein expression and purification was identical to Prp24 with a few exceptions: the expressed protein had a TEV-labile polyhistidine tag that was removed after IMAC purification, dialysis was into a modified ion exchange buffer (100 mM NaCl, 10 mM tris, 10 mM tris-HCl, 20% glycerol, 1 mM TCEP-HCl, 1 mM MnCl₂, pH ~8.0) and a 1 mL HiTrapQ column (GE Healthcare) was used instead of a heparin column in the above buffer without manganese.

**Mutant AtRNL protein expression and purification**. A mutant variant of *Arabidopsis thaliana* protein AtRNL containing only the kinase and 2′,3′-cyclic phosphate 3′-phosphodiesterase domains (residues 677–1104)[31,54,55] was used to prepare U6 nucleotides 30–112 with a 2′ non-cyclic phosphate (see below). A modified pET vector with a TEV-labile polyhistidine tag was used for protein expression and purification, using the same approach as above for Prp24, except that 1 mg TEV protease was added during dialysis and a 5 mL HiTrapSP column (GE Healthcare) was used instead of a heparin column during ion exchange purification.

**RNA production for crystallization**. In vitro transcription with T7 RNA polymerase was used to synthesize *S. cerevisiae* U6 nucleotides 30–112 or 30–113 from a linearized pUC57 plasmid template harboring a T7 promoter (TTCTAA-TACGACTCACTATA) and a minimal 56 nucleotide HDV ribozyme "drz-Mtgn-3"[56] to ensure a homogenous 3′ end in U6 with a cyclic phosphate that could subsequently be converted to a non-cyclic phosphate after purification of U6 (see below). The 30–112 nucleotide RNA contained an A62G mutation in U6, and the 30–113 nucleotide RNA contained an A62G/A79G double mutation. The transcription reaction contained ~0.1–0.5 μM linearized plasmid template, ~0.5 mg mL⁻¹ T7 RNA polymerase, 5 mM each of ATP, GTP, CTP, and UTP; 100 mM tris, 100 mM tris-HCl, 40 mM MgCl₂, 5 mM DTT, 1 mM spermidine trihydrochloride and 0.01% (v/v) Triton X-100. The transcription reaction was performed at 37 °C for 1 h and halted by addition of 100 mM trisodium EDTA and 20% glycerol. The RNA was purified via denaturing 10% polyacrylamide gel electrophoresis in 8 M urea and 100 mM tris, 100 mM boric acid, 1 mM EDTA. The RNA was identified by UV shadowing, extracted by scalpel, and removed from the gel matrix by passive diffusion overnight at room temperature into a solution containing 300 mM sodium acetate, 50 mM HCl, 1 mM EDTA, 1 mM sodium azide, pH ~5.6. Soluble RNA was separated from solid acrylamide by filtration and then applied to a 5 mL HiTrap Q column (GE Healthcare) that had been equilibrated in RNA chromatography buffer (300 mM NaCl, 10 mM KH₂PO₄, 10 mM K₂HPO₄, 1 mM EDTA, 1 mM sodium azide). Bound RNA was

washed with 20 mL of buffer and then eluted with buffer adjusted to 2 M NaCl. RNA was precipitated by addition of one volume 100% isopropanol, pelleted, washed with 70 % ethanol, and then resuspended in 100 mM KCl, 20 mM bis-tris, 10 mM HCl, 1 mM EDTA, 1 mM sodium azide, pH ~6.5. The RNA concentration was estimated using UV absorption and anticipated extinction coefficients at 260 nm[57].

**3′ end modification of RNA.** *Escherichia coli* YfcE(C74H) was used to convert the 2′,3′ cyclic phosphate moiety in U6 to a 3′ non-cyclic phosphate[53] in a reaction mixture containing 20 μM RNA, 2 μM YfcE, 100 mM KCl, 20 mM bis-tris, 10 mM HCl, 1 mM MnCl$_2$, and pH ~6.5. AtRNL (residues 677–1104) was used to convert the cyclic phosphate moiety in U6 to a 2′ non-cyclic phosphate[31,54,55] in a reaction mixture containing 30 μM RNA, 3 μM AtRNL, 100 mM KCl, 20 mM bis-tris, 10 mM HCl, 1 mM EDTA, and pH ~6.5. Reactions were incubated at 37 °C for 1 h prior to direct application of reaction mixtures to 1 mL HiTrapQ columns (GE Healthcare) and ion exchange purification as above for RNA after in vitro transcription. The ability of calf intestinal phosphatase (New England Biolabs) treatment to confer aberrant mobility on denaturing tris-borate gels was used as an assay to confirm complete conversion of the terminal cyclic phosphate to a non-cyclic phosphate[37].

**Reconstitution of U6 snRNPs.** Non-denaturing polyacrylamide gel electrophoresis was used to determine optimal ratios of protein and RNA prior to in vitro snRNP reconstitution, in order to correct for errors in absolute concentration of each component and prevent super-stoichiometric association of protein and RNA. Subsequently, snRNPs were reconstituted at final concentrations of 10–100 μM in a final volume of ~1 mL and dialyzed overnight at 4 °C against 1 L of buffer containing either 100 mM KCl, 10 mM HEPES acid, 10 mM sodium HEPES, 1 M TCEP-HCl, 1 M sodium azide, 0.1 mM EDTA, and pH ~7.0 (for the 2′ phosphate-terminated snRNP) or 50 mM KCl, 50 mM ammonium fluoride, 10 mM HEPES acid, 10 mM sodium HEPES, 2 mM magnesium sulfate, 1 mM TCEP-HCl, 1 mM sodium azide, and pH ~7.0 (for the 3′ phosphate-terminated snRNP). Samples were then concentrated with 50 kDa spin filters to a final snRNP concentration of 5–20 mg mL$^{-1}$. The 2′ phosphate-terminated snRNP was further purified by Superdex 200 10/300 (GE Healthcare) gel filtration chromatography in fresh dialysis buffer prior to concentration and crystallization.

**Crystallization of U6 snRNP with a 2′ phosphate.** U6 snRNP containing U6 nucleotides 30–112 with an A62G mutation and a terminal 2′ phosphate, Prp24 residues 1–444, and Lsm2–8 corresponding to the "mostly wild type" construct[38] was crystallized by hanging drop vapor diffusion at 16 °C. 2 μL of snRNP was mixed with 2 μL of crystallization reagent containing 50 mM HEPES acid, 50 mM sodium HEPES, 200 mM ammonium fluoride, 18% PEG 3350, and 10 mM magnesium chloride. Crystals were briefly transferred to a cryogenic solution (50 mM HEPES acid, 50 mM sodium HEPES, 200 mM ammonium fluoride, 18% PEG 3350, 10 mM magnesium chloride, 15% glycerol, 100 mM potassium chloride, 1 mM TCEP-HCl, 1 mM sodium azide, and 0.1 mM trisodium EDTA) prior to vitrification in liquid nitrogen. Diffraction data were collected at 100 K from a single crystal on beamline 24-ID-C at the Advanced Photon Source. The final diffraction dataset was obtained by merging two datasets collected from two different regions of the same crystal.

**Crystallization of U6 snRNP with a 3′ phosphate.** U6 snRNP containing U6 nucleotides 30–113 with A62G and A79G point mutations and a terminal 3′ phosphate, Prp24 residues 27–444, and the modified Lsm2–8 ring described above was crystallized by hanging drop vapor diffusion at 16 °C. A volume of 1 μL of snRNP was mixed with 1 μL of crystallization reagent containing 300 mM sodium potassium tartrate, 17.5% PEG 3350, 17–22% glycerol, and 2 mM manganese(II) chloride. Crystals were vitrified by direct immersion into liquid nitrogen. Anisotropic diffraction data were collected at 100 K from three crystals on beamline 21-ID-D at the Advanced Photon Source. The final diffraction dataset was obtained by merging nine distinct datasets.

**Crystal structure determination.** Diffraction data were integrated using XDS[58]. Space group determination and scaling/merging were performed in *POINTLESS*[59] and *AIMLESS*[60], respectively. STARANISO[61] was subsequently used for ellipsoidal truncation of the highly anisotropic diffraction data obtained from the 3′ phosphate U6 snRNP crystals in space group *P2$_1$*, which resulted in improved electron density in unfilled 2m$F_o$−D$F_c$ maps after molecular replacement and refinement. *Phenix.xtriage* was used to assay potential twinning in the diffraction data[62]. Initial phases were determined by molecular replacement using *Phaser*[63] with initial search templates PDB 4N0T (U6/Prp24 core)[22], 4M7D (Lsm8)[38], and 4C92 (Lsm2–7)[44]. Structure refinement was performed in *Phenix.refine*[62,64] using secondary structure restraints, reference model restraints, and TLS parameterization, with iterative rounds of manual model building in Coot[65,66] and additional automated refinement in *Phenix.refine*, ERRASER[67] and *phenix.rosetta_refine*[68]. Reference model restraints were essential to refinement of the Lsm2–8 ring in PDB 6ASO, due to apparent dynamics in the ring and resulting poor density for Lsm2,

Lsm4, Lsm7, and Lsm8, which is most pronounced near to the periphery of the ring. Density for the C-terminal region of Prp24 (near to Lsm5 and Lsm7) was most compelling in PDB 5VSU, and the corresponding region of Prp24 was not included in the final model for PDB 6ASO. Data collection and refinement statistics are given in Supplementary Table 1. The electrostatic surface was calculated using APBS[69] as implemented in PyMOL. All figures were generated with PyMOL (http://www.pymol.org) and Coot[66]. A stereo image of representative density is shown in Supplementary Fig. 6. Surface conservation in the Lsm2–8 ring depicted in Supplementary Fig. 5 was generated with the alignment in Supplementary Note 1 and Consurf[70].

**RNA production for in vitro binding and annealing analysis.** The small 5′-FAM labeled RNA used for fluorescence polarization (*S. cerevisiae* U6 nucleotides 104–113) was purchased from Integrated DNA Technologies and purified by urea PAGE and ion exchange as above, omitting the alcohol precipitation step in favor of iterative dilution/re-concentration with 3 kDa spin concentration filters (EMD Millipore) to exchange buffer. Calf intestinal alkaline phosphatase (CIP) (New England Biolabs) was used to remove the 3′ terminal phosphate[71].

In vitro transcribed RNAs used in Supplementary Fig. 3 were radiolabeled. RNAs were dephosphorylated with CIP (New England Biolabs), extracted with 25:24:1 phenol:chloroform:isoamyl alcohol and ethanol precipitated prior to 5′ end labeling with T4 polynucleotide kinase (PNK) (New England Biolabs) and 3000 Ci mmol$^{-1}$ [γ−$^{32}$P]-ATP (Perkin Elmer). RNAs were purified by urea PAGE, extracted from the gel by passive diffusion into 300 mM sodium acetate pH 5.2, 1 mM EDTA, and ethanol precipitated.

The gels in Fig. 3d, e used a 5′-Cy3-labeled RNA containing U6 nucleotides 1–112 that was produced via splinted ligation using a 5′-Cy3 U6 1–12 RNA oligonucleotide (Integrated DNA Technologies) and U6 13–112 that was produced via in vitro transcription from a plasmid containing a 5′ hammerhead ribozyme and a 3′ HDV ribozyme. The U6 13–112 RNA was phosphorylated with ATP and T4 PNK, which also removed the 2′,3′ cyclic phosphate produced by 3′ HDV cleavage. The Cy3-U6 1–12 and U6 13–112 RNAs were ligated using T4 RNA ligase 2 (New England Biolabs) at 37 °C for 2 h using a DNA splint that was complementary to U6 nucleotides 1–30. The ligation product was purified by urea PAGE and ion exchange as above.

Full-length 3′-Cy5 fluorescently labeled U4 (nucleotides 1–160) used in Fig. 3f was produced via splinted ligation as described above using a synthetic RNA oligonucleotide consisting of U4 146–160 with a 3′-Cy5 moiety, in vitro transcribed U4 1–145 that was produced from a pUC57 plasmid containing a 5′ hammerhead and a 3′ HDV ribozyme, and a DNA oligonucleotide complementary to nucleotides 120–160 of U4. U4 1–145 was treated with T4 PNK in the absence of ATP to remove the 2′,3′ cyclic phosphate and the U4 145–160-Cy3 RNA was 5′ phosphorylated with T4 PNK in the presence of ATP as described above. After splinted ligation, the resulting U4 1–160-Cy5 was purified by urea PAGE and ion exchange as described above.

**Electrophoretic mobility shift assays.** Binding of RNAs with Prp24 and Lsm2–8 was performed essentially as described elsewhere[24]. Trace (<1 nM) 5′ $^{32}$P-labeled RNA or 0.5 nM Cy3-U6 1–112 were heated to 90 °C for 2 min in 2 × RNA binding buffer (100 mM KCl, 20% sucrose, 20 mM bis-tris, 10 mM HCl, 1 mM EDTA acid, 1 mM TCEP-HCl, 0.01% Triton X-100, pH ~6.5, 0.2 mg mL$^{-1}$ yeast tRNA, and 0.02 mg mL$^{-1}$ sodium heparin) and snap cooled on wet ice. Proteins were prepared as 2 × stocks in protein binding buffer (100 mM KCl, 20% sucrose, 20 mM bis-tris, 10 mM HCl, 1 mM EDTA, 1 mM TCEP-HCl, 0.01% Triton X-100, pH ~6.5, and 0.2 mg mL$^{-1}$ BSA). Binding reactions were prepared at room temperature containing equal volumes of 2 × RNA and protein stocks. For the U6 snRNP binding gels, different amounts of Lsm2–8 were added to trace U6 that had been pre-incubated for 20 min with 20 nM Prp24. Final samples were incubated at room temperature for 20 min prior to loading onto 16.5 × 22 × 0.15 cm 6% poly-acrylamide gels (29:1 acrylamide:bis-acrylamide, 89 mM tris borate, and 2 mM EDTA pH 8.0). Samples were electrophoresed for 2–3 h at 150 V at 4 °C. Radioactive gels were dried on filter paper, exposed to a PhosphorImager screen, and imaged on a Typhoon FLA 9000 biomolecular imager (GE Healthcare Life Sciences). Fluorescent gels were imaged directly through low fluorescence glass plates (CBS Scientific) using a Typhoon FLA 9000 imager. Results were analyzed using ImageJ software and binding curves were fit using nonlinear regression in GraphPad Prism 4 to the Hill equation: % bound = $(B_{max} \times [Prp24]^H)/(K_d^H + [Prp24]H)$. $B_{max}$ was restrained to be between 0 and 100%, and the $H$ (Hill coefficient) and $K_d$ were restrained to be >0. Binding affinities are reported for the average of three technical replicates. Uncropped gel images are shown in Supplementary Fig. 7.

**In vitro annealing assays.** Annealing reactions were carried out essentially as described elsewhere[24] at 22 °C in binding buffer (as described above) containing 2 nM U4 1-160-Cy5, 10 nM U6 RNA, and 200 nM Prp24 protein or 50 nM Prp24/50 nM Lsm2–8. Reactions were stopped by the addition of 2 μl of proteinase K buffer (0.5% sodium dodecyl sulfate, 0.3 mg mL$^{-1}$ tRNA, 5 mM CaCl$_2$, 30 mM HEPES pH 7.0, and 0.2 mg mL$^{-1}$ proteinase K) prior to separation on a 6%

polyacrylamide gel (29:1 acrylamide:bis-acrylamide, 89 mM tris borate, 2 mM EDTA pH 8.0). Samples were electrophoresed for 2 h at 3 W at 4 °C, and analyzed as above. Annealing rates were calculated using the ratio of free U4 to U4/U6 in Proteinase K-treated lanes. Resulting data were then fit to a one-phase exponential association equation (GraphPad Prism 4). Annealing rates are reported for three technical replicates. Uncropped gel images are shown in Supplementary Fig. 7.

**In vivo analysis of Prp24 mutants**. Mutations in pRS313-ScPrp24[72] were introduced via inverse PCR with Phusion DNA polymerase (New England Biolabs), treated with DpnI, and self-ligated using T4 PNK and T4 DNA ligase (New England Biolabs) prior to transformation into *E. coli* NEB 5α competent cells (New England Biolabs), plasmid isolation, and Sanger sequencing. Mutant alleles of *PRP24* in pRS313 were transformed into LL101[72] (*MAT*a *his3 leu2 trp1 ura3 met2 can1 ade2 lys2 prp24-Δ1::ADE2* [pUN50-PRP24]) via the lithium acetate method[73] and selected on solid -his media. Transformants were restreaked onto -his media containing 1 mg mL$^{-1}$ 5-fluoroorotic acid to select against the wild-type plasmid, then restreaked back onto -his media. Viable clones were grown in YEPD liquid media overnight, then diluted in 10% glycerol to an $OD_{600}$ of 0.5. Serial ten-fold dilutions were plated on solid YEPD media and incubated at 16, 23, 30, and 37 °C. Doubling times were measured in liquid YEPD media by taking a saturated overnight culture, diluting to an $OD_{600} < 0.2$ with growth at 30 °C. The $OD_{600}$ of the culture was measured every 30 min. The slope of the log-linear portion of the curve ($OD_{600}$ 0.2–0.8) was fit using linear regression in Excel and the doubling time was calculated using the formula doubling time = log(2)/slope. The doubling time at 37 °C was determined by growing the cultures at 30 °C until they reached log phase growth, then diluting with an equal volume of 44 °C YEPD to quickly shift the cultures to 37 °C.

**In vivo analysis of Lsm mutants**. PCR was used to amplify genomic loci for Lsm2, Lsm3, and Lsm8 from *S. cerevisiae* strain BY4705 (*MAT*α, *ade2*Δ::*hisG*, *his3*Δ200, *leu2*Δ0, *lys2*Δ0, *met15*Δ0, *trp1*Δ63, and *ura3*Δ0)[74] with flanking BamHI and XhoI restriction enzyme sites to allow subsequent cloning of the PCR products into plasmids pRS414 and pRS416. The Lsm2 amplicon contained 344 bp upstream and 433 bp downstream of the Lsm2 ORF. For unknown reasons, the Lsm2 intron greatly impeded plasmid propagation in *E. coli* and was therefore removed from the pRS414-Lsm2 plasmid prior to site directed mutagenesis and plasmid shuffle of mutant alleles in vivo. The Lsm3 amplicon contained 313 bp upstream and 134 bp downstream of the Lsm3 ORF. The Lsm8 amplicon contained 205 bp upstream and 133 bp downstream of the Lsm8 ORF.

pRS416 plasmids containing wild-type alleles of Lsm genes were transformed into strain BY4705 with subsequent selection on synthetic defined –ura media. The resulting strains were transformed with ~1 μg of DNA encoding the kanMX cassette[75] flanked by 60 bp immediately upstream and downstream of the genomic Lsm2, Lsm3, or Lsm8 ORFs, with subsequent selection on YEPD plates containing 500 μg mL$^{-1}$ G418. Transformants were restreaked onto fresh YEPD/G418 plates, then inoculated into YEPD liquid culture and grown to saturation before preparing freezer stocks for each strain. The genomic disruptions were confirmed by PCR amplification and Sanger sequencing of the disrupted genomic loci. Strain EJM001 corresponds to BY4705 *lsm2*Δ::*kanMX* pRS416-ScLsm2, strain EJM002 corresponds to BY4705 *lsm3*Δ::*kanMX* pRS416-ScLsm3, and strain EJM003 corresponds to BY4705 *lsm8*Δ::*kanMX* pRS416-ScLsm8. *LSM* alleles in pRS414 were shuffled into the respective disruption strain by transformation and selection on –trp media, followed by overnight growth in –trp liquid media prior to counter selection of the pRS416-borne wild-type Lsm alleles on medium containing 1 mg mL$^{-1}$ 5-fluoroorotic acid at 30 °C. Viable clones were grown in YEPD liquid medium to an $OD_{600}$ of ~0.5, pelleted and resuspended in fresh YEPD to normalize the $OD_{600}$ to 1.0, and then spotted to YEPD plates using a 48-pin inoculation manifold and serial ten-fold dilutions. The resulting plates were incubated at 16, 23, 30, and 37 °C.

**Fluorescence polarization**. Fluorescence polarization binding assays were performed by mixing 100 μL of 2 × RNA buffer (0.6 nM FAM-labeled RNA, 100 mM KCl, 10 mM HEPES acid, 10 mM sodium HEPES, 10% glycerol, 1 mM TCEP-HCl, 0.2 mg mL$^{-1}$ tRNA (Roche), and 0.02 mg mL$^{-1}$ sodium heparin (Sigma)) and 100 μL of Lsm2–8 at varying concentration in 2 × protein buffer (100 mM KCl, 10 mM HEPES acid, 10 mM sodium HEPES, 10% glycerol, 1 mM TCEP-HCl, and 0.2 mg mL$^{-1}$ BSA (Ambion)) in black 96 well microplates (Greiner Bio-One). The Lsm2-K20E and Lsm3-R21E titrations contained the same total amount of probe in each well, but were at half the above volumes with twice the final probe concentration. Fluorescence polarization was measured on a Tecan Infinite M1000Pro using an excitation wavelength of 470 nm and emission wavelength of 519 nm. Gain was optimized for each microplate. Fluorescence polarization was measured in duplicate from two independent titrations. Binding curves were fit using nonlinear regression in GraphPad Prism 4 to the following four parameter equation: $FP = FP_{min} + (FP_{max} - FP_{min})/(1 + 10^{((\log K_d - \log[\text{protein}])*H)})$, where $FP_{min}$ and $FP_{max}$ are the minimum and maximum polarizations, $K_d$ is the binding dissociation constant, and $H$ is the Hill coefficient. H was constrained to be one during nonlinear regression. Depicted binding curves are normalized to $FP_{min}$ and $FP_{max}$.

**Data availability**. Coordinates and structure factors have been deposited in the Protein Data Bank with accession codes 5VSU and 6ASO. Other data supporting the findings of this manuscript are available from the corresponding author upon request.

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

## Acknowledgements

We are grateful to A. Hoskins and members of the Brow and Butcher laboratories for helpful discussions and critical reading of the manuscript; J. Schuermann, K. Perry, and K. Rajashankar for assistance with collection of diffraction data; R. Frederick for assistance with protein production; K. Nagai for the gift of recombinant Lsm2–8 in the early stages of this project; and Y. Shi for the pQLink-Lsm2–8 expression system. Use of the Advanced Photon Source, an Office of Science User Facility operated for the U.S. Department of Energy (DOE) Office of Science by Argonne National Laboratory, was supported by the U.S. DOE under Contract No. DE-AC02-06CH11357. Use of NE-CAT was supported by the National Institutes of Health (NIH) grants P41 GM103403 and S10 RR029205. Use of LS-CAT was supported by NIH grant 085P1000817. S.E.B was supported by the NIH grants R01 GM065166 and R35 GM118131. D.A.B was supported by NIH grants R01 GM065166 and R35 GM118075. A.L.D. was supported by the University of Wisconsin-Madison Louis and Elsa Thomsen Wisconsin Distinguished Graduate Fellowship. Fluorescence polarization data were obtained at the University of Wisconsin-

Madison Biophysics Instrumentation Facility, which was established with support from the University of Wisconsin-Madison and grants BIR-9512577 (NSF) and S10RR13790 (NIH).

## Author contributions

E.J.M., A.L.D., G.K.S., and A.D.Y. prepared reagents. E.J.M. and A.D.Y. collected diffraction data. E.J.M. determined the crystal structures. E.J.M. and A.L.D. performed yeast work. A.L.D. performed in vitro binding and annealing assays. E.J.M., D.A.B., and S.E.B. supervised the work and wrote the manuscript.

## Additional information

**Competing interests:** The authors declare no competing interests.

