## [Peer Review File · Nature Communications]

REVIEWERS' COMMENTS:

Reviewer #1 (Remarks to the Author):

The manuscript by Montemayor et al. describes the crystal structure of the U6 snRNP (Prp24 and the Lsm2-8 complex bound to an almost full-length U6 snRNA). The structure of the Prp24-U6 snRNA domain is nearly identical to the structure of this complex determined in isolation and the current structure provides additional information regarding the interaction between the Prp24 and the Lsm2-8 complex. Prp24 interacts with the Lsm2-8 domain mainly through the interaction between the RRM4 domain of Prp24 and Lsm2 but also through contacts between the conserved C-terminus of Prp24 and Lsm5 and Lsm7. So the obvious question is if the observed contacts are biologically important or artefact of crystal packing. In this rather short manuscript the authors presented high quality biochemical and genetic data to support the importance of this interaction. The manuscript also present interesting data regarding the interaction between the 3' end U6 snRNA and the Lsm2-8 ring. I am happy to support the publication of this manuscript in Nature Communication but the answer to the following question would make this manuscript more interesting.

Is the structure of the Prp24-U6 snRNA in the context of the complete U6 snRNP identical to the Prp24-U6 snRNA complex in isolation? It would be helpful to provide rmsd.

Eight RRM4 residues at the interface with alanine ("Prp24-8Asub") and assayed U6 and Lsm2-8 binding affinity in vitro. Prp24-8Asub had unaltered affinity for U6 snRNA (Fig. 2d), but reduced the affinity of U6•Prp24 for Lsm2-8 two-fold. Isn't this surprising small difference if this interaction is important?

If Prp24 interacts with the Lsm2-8 mainly through Lsm2 why do we not find Prp24-U6 snRNA-Lsm1-7 complex in vivo. Why is the Lsm1-7 complex cytoplasmic and the Lsm2-8 complex nuclear?

Reviewer #2 (Remarks to the Author):

U6 snRNA plays a central catalytic role in the spliceosome and it undergoes multiple conformational rearrangements in its life cycle to enable its catalytic function. The authors determined the crystal structure of the U6 snRNP including U6 snRNA, Prp24, and seven Lsm proteins. Other than some small truncated or disordered regions, the structure contains the complete U6 snRNP. The paper is well written and provided novel structural and functional insight into the complete U6 snRNP and how it may ultimately help U6 snRNA to achieve its function in the spliceosome. More specifically, it reveals how the Prp24 and Lsm protein-protein interactions help placing the Lsm ring in close proximity to the "active site" of Prp24, providing the structural basis of how the Lsm ring can facilitate U4/U6 annealing. The structure also reveals how the Lsm ring specifically recognizes modified U6 snRNA which potentially promotes the selective recruitment of mature U6 snRNA into the spliceosome. There are a number of minor issues in the manuscript that need to be addressed.

1. It is not clear through the entire main text of the manuscript whether "the structure" is a crystallographic or cryo EM structure. This should be clarified the first time that the paper mentions "the structure". It should also briefly describe how the complex was obtained, through reconstitution or purified from natural source, early in the text.

2. There is no background on Lsm1 and it is not clear what is the functional significance on the observed homology between Lsm8 and Lsm1.

4. Line 40 seems to refer to both residues that are not in the constructs and residues that are disordered, but it does not have a complete list of residues that are disordered (such as those grey residues in Fig. 1a and additional disordered residues in Prp24).
5. Line 43, the claim that the U6 snRNP structure “represents the most complete view of an individual snRNP subunit architecture to date” does not add anything to the paper and is not particularly meaningful. U6 snRNP is probably the simplest snRNP.
6. Line 81, can the authors elaborate on why the binding pocket observed in this structure is different from what is previously observed, and the physiological relevance of each.
7. Line 116, Extended Data Figure 8 should be 7.
8. Line 119, the authors should define “proximal” face and label it in Fig. 1c.
9. Extended data table 1. There seems to be quite a number of Ramachandran outliers. At this resolution, there really are not good justifications or confidence that these are true outliers. The authors should remodel these residues to remove outliers.
10. Extended Data File 1 was cited in line 39 but does not seem to exist.

Response to referee comments

Comments from referee # 1:

The manuscript by Montemayor et al. describes the crystal structure of the U6 snRNP (Prp24 and the Lsm2-8 complex bound to an almost full-length U6 snRNA). The structure of the Prp24-U6 snRNA domain is nearly identical to the structure of this complex determined in isolation and the current structure provides additional information regarding the interaction between the Prp24 and the Lsm2-8 complex. Prp24 interacts with the Lsm2-8 domain mainly through the interaction between the RRM4 domain of Prp24 and Lsm2 but also through contacts between the conserved C-terminus of Prp24 and Lsm5 and Lsm7. So the obvious question is if the observed contacts are biologically important or artefact of crystal packing. In this rather short manuscript the authors presented high quality biochemical and genetic data to support the importance of this interaction. The manuscript also present interesting data regarding the interaction between the 3'end U6 snRNA and the Lsm2-8 ring. I am happy to support the publication of this manuscript in Nature Communication but the answer to the following question would make this manuscript more interesting.

Is the structure of the Prp24-U6 snRNA in the context of the complete U6 snRNP identical to the Prp24-U6 snRNA complex in isolation? It would be helpful to provide rmsd.

We have added the requested comparison to the main text:

“The core region of the U6 snRNP retains the unique “interlocked rings” topology of Prp24 and U6 snRNA and is virtually identical to previously determined structures of U6/Prp24 binary complexes^{22,23}, with a comparative r.m.s.d of approximately 1.3 Å.”

Eight RRM4 residues at the interface with alanine (“Prp24-8Asub”) and assayed U6 and Lsm2-8 binding affinity in vitro. Prp24-8Asub had unaltered affinity for U6 snRNA (Fig. 2d), but reduced the affinity of U6•Prp24 for Lsm2-8 two-fold. Isn't this surprising small difference if this interaction is important?

The binding defect is indeed small, but we believe it is important nevertheless since it is qualitatively mirrored by reduced *in vitro* annealing rates and yields a detectable growth phenotype *in vivo* (Figs. 3f,g). We added the following sentence to the end of the relevant results section:

“Together, these findings show that the observed architecture of the U6 snRNP likely represents an on-path species in Prp24-mediated annealing of U4/U6 di-snRNPs, where placement of the Lsm2-8 ring in close proximity to the electropositive groove of Prp24 is partly, but not entirely, responsible for enhanced annealing in the presence of Lsm2-8”

If Prp24 interacts with the Lsm2-8 mainly through Lsm2 why do we not find Prp24-U6 snRNA-Lsm1-7 complex in vivo. Why is the Lsm1-7 complex cytoplasmic and the Lsm2-8 complex nuclear?

We have answered this question by adding the following sentences to the end of the relevant results section:

“The observed protein-protein contacts in the U6 snRNP are compatible with binding of either the Lsm1-7 or Lsm2-8 rings in U6 snRNPs. However, Lsm1 lacks the non-consensus nuclear localization sequence found in Lsm8 (ref. 36), and therefore cytoplasmic Lsm1-7 rings are unlikely to be recruited into U6 snRNPs.”

Comments from referee # 2:

U6 snRNA plays a central catalytic role in the spliceosome and it undergoes multiple conformational rearrangements in its life cycle to enable its catalytic function. The authors determined the crystal structure of the U6 snRNP including U6 snRNA, Prp24, and seven Lsm proteins. Other than some small truncated or disordered regions, the structure contains the complete U6 snRNP. The paper is well written and provided novel structural and functional insight into the complete U6 snRNP and how it may ultimately help U6 snRNA to achieve its function in the spliceosome. More specifically, it reveals how the Prp24 and Lsm protein-protein interactions help placing the Lsm ring in close proximity to the “active site” of Prp24, providing the structural basis of how the Lsm ring can facilitate U4/U6 annealing. The structure also reveals how the Lsm ring specifically recognizes modified U6 snRNA which potentially promotes the selective recruitment of mature U6 snRNA into the spliceosome.

There are a number of minor issues in the manuscript that need to be addressed.

1. It is not clear through the entire main text of the manuscript whether “the structure” is a crystallographic or cryo EM structure. This should be clarified the first time that the paper mentions “the structure”. It should also briefly describe how the complex was obtained, through reconstitution or purified from natural source, early in the text.

This is now stated in the title and first sentence of the first paragraph in the results section:

*“Crystal structure of the U6 snRNP. We reconstituted the U6 snRNP from *S. cerevisiae*, using in vitro transcribed U6 snRNA and recombinant Prp24 and Lsm2–8.”*

2. There is no background on Lsm1 and it is not clear what is the functional significance on the observed homology between Lsm8 and Lsm1.

We have added a figure to the main text that shows a direct comparison of the Lsm1 -7 and Lsm2-8 structures. We also added the following paragraph to the discussion section:

“Free Lsm1-7 lacks robust binding affinity for RNA despite pronounced structural similarity between the nucleotide binding pockets of Lsm1 and Lsm8 (ref. 38,43,44) (Fig. 4e). In light of the important role of the C-terminus of Lsm8 observed here, we hypothesize that the C-terminal region of Lsm1 (Fig. 2d) inhibits binding of RNA to Lsm1-7. Since association of Pat1 with Lsm1-7 restores binding affinity for RNA⁵⁰, we further speculate that Pat1 acts as an allosteric regulator of Lsm1-7, stimulating RNA binding activity via displacement of the Lsm1 C-terminus from the distal face of the ring. Further work is required to identify the exact sequence of RNA that associates with Lsm1-7 and the corresponding structure of the Lsm1.”

4. Line 40 seems to refer to both residues that are not in the constructs and residues that are disordered, but it does not have a complete list of residues that are disordered (such as those grey residues in Fig. 1a and additional disordered residues in Prp24).

The disordered Lsm residues are annotated in in Supplementary Note 1. We added a comment to the results section stating that the linker between RRM4 and the SNFFL box is disordered in Prp24, and the exact residues are now listed in the legend for Figure 1.

5. Line 43, the claim that the U6 snRNP structure “represents the most complete view of an individual snRNP subunit architecture to date” does not add anything to the paper and is not particularly meaningful. U6 snRNP is probably the simplest snRNP.

The sentence in question has been removed from the manuscript.

6. Line 81, can the authors elaborate on why the binding pocket observed in this structure is different from what is previously observed, and the physiological relevance of each.

We now state in the results section:

“Instead of being bound in the Sm-like pocket of Lsm3, the terminal nucleotide occupies a new pocket comprised of residues in Lsm2, Lsm3 and Lsm8 (Fig. 4d,e), and the penultimate nucleotide is bound in the previously identified Sm-like pocket of Lsm3.”

The physiological relevance of this new binding pocket follows the mutational data in the results section:

“... the above mutations invert Lsm2-8’s binding preference for 3’-end processed U6 snRNA, with the mutants preferentially binding to a non-phosphorylated 3’ end (Fig. 5 and Table 3). Thus, the 3’ binding pocket observed here defines the structural basis for preferential incorporation of 3’-end processed U6 snRNA into spliceosomes.”

7. Line 116, Extended Data Figure 8 should be 7.

Fixed.

8. Line 119, the authors should define “proximal” face and label it in Fig. 1c.

The “proximal”, “distal” and “rim” nomenclatures are now cited in the introduction section:

“...homology between the Lsm ring and the Hfq protein families²⁸, which bind and remodel RNAs through numerous “proximal”, “distal” and “rim” binding sites that place cognate RNAs in close proximity to one another^{29,30}.”

The requested annotation has been added to Figure 1c, and is also extensively annotated in Supplementary Figure 5.

9. Extended data table 1. There seems to be quite a number of Ramachandran outliers. At this resolution, there really are not good justifications or confidence that these are true outliers. The authors should remodel these residues to remove outliers.

We have gone through the models and fixed any obvious errors. The final models have Ramachandran statistics that are near to the median of all reported structures at their respective resolutions:

10. Extended Data File 1 was cited in line 39 but does not seem to exist.

This appears to be a data transfer issue with the journal that will be fixed during resubmission.